# The Use of Multilayer Perceptron Artificial Neural Networks to Detect Dairy Cows at Risk of Ketosis

**DOI:** 10.3390/ani12030332

**Published:** 2022-01-29

**Authors:** Edyta A. Bauer, Wojciech Jagusiak

**Affiliations:** 1Department of Animal Reproduction, Anatomy and Genomics, Faculty of Animal Science, University of Agriculture in Krakow, 30-059 Kraków, Poland; 2Department of Genetics, Animal Breeding and Ethology, Faculty of Animal Science, University of Agriculture in Krakow, 30-059 Kraków, Poland; wojciech.jagusiak@urk.edu.pl

**Keywords:** dairy cattle, ketosis, multi-layer perceptron, practical application

## Abstract

**Simple Summary:**

Ketosis is a serious metabolic disease in high-yield dairy cows, that affects productive herds throughout the world. Subclinical ketosis is one of the most dominant metabolic disorders in dairy herds during early lactation, so early detection and prevention are important for both economic and animal welfare reasons. Neural networks, which offer a high degree of accuracy in predicting various phenomena and processes where there is no clear causal correlation or there are no rules that allow the establishment of a logical cause-and-effect relationship, can be used to address problems related to prediction, classification, or control. A Multi-Layer perceptron (MLP) is a feedforward artificial neural network model that takes input data for a set of proper output. This study investigated the performance of four algorithms used to train MLP networks. The experimental results demonstrate that the MLP network model improved the accuracy of process recognition of subclinical ketosis in dairy cows. The received artificial model’s results were saved in the predictive model markup language (PMML) and can be used to describe the learning set, the algorithm used in the data mining application and related information.

**Abstract:**

Subclinical ketosis is one of the most dominant metabolic disorders in dairy herds during lactation. Cows suffering from ketosis experience elevated ketone body levels in blood and milk, including β-hydroxybutyric acid (BHB), acetone (ACE) and acetoacetic acid. Ketosis causes serious financial losses to dairy cattle breeders and milk producers due to the costs of diagnosis and management as well as animal welfare reasons. Recent years have seen a growing interest in the use of artificial neural networks (ANNs) in various fields of science. ANNs offer a modeling method that enables the mapping of highly complex functional relationships. The purpose of this study was to determine the relationship between milk composition and blood BHB levels associated with subclinical ketosis in dairy cows, using feedforward multilayer perceptron (MLP) artificial neural networks. The results were verified based on the estimated sensitivity and specificity of selected network models, an optimum cut-off point was identified for the receiver operating characteristic (ROC) curve and the area under the ROC curve (AUC). The study demonstrated that BHB, ACE and lactose (LAC) levels, as well as the fat-to-protein ratio in milk, were important input variables in the network training process. For the identification of cows at risk of subclinical ketosis, variables such as BHB and ACE levels in milk were of particular relevance, with a sensitivity and specificity of 0.84 and 0.61, respectively. It was found that the back propagation algorithm offers opportunities to integrate artificial intelligence and dairy cattle welfare within a computerized decision support tool.

## 1. Introduction

Ketosis is one of the most common medical conditions in dairy cattle [1,2]. Intensive selection combined with new feeding systems have resulted in a significant increase in the performance of dairy cows [3], but high-yield milk production may be associated with negative energy balance (NEB) and ketosis, which is one of the most serious metabolic diseases affecting dairy cow herds across the world [2,4,5]. Ketosis causes major economic losses attributable to reduced milk yield, reduced health status, impaired reproductive performance, and high rates of culling of affected cows [6,7,8]. The diseases that have been reported are milk fever, displaced abomasum, mastitis, and metritis [1,6,7,8]. In early lactation, dairy cows typically have an NEB, which has been related to metabolic disorders such as subclinical ketosis [2,4,5,9]. Ketosis is mostly observed during the initial stage of lactation and can be diagnosed based on elevated levels of ketone bodies (β-hydroxybutyric acid (BHB), acetone (ACE), and acetoacetic acid) in milk, urine and blood [2,10,11,12]. Ketosis is a metabolic disease of heterogeneous etiology, which is usually subclassified into primary ketosis, observed during the postpartum period, and secondary ketosis, attributable to the existence of other medical conditions, as well as alimentary ketosis, which is associated with feeding problems (NEB) during the postpartum period and early lactation [2,11,12,13,14,15,16]. Primary ketosis attributable to NEB in the postpartum period is further subdivided into type 1 primary ketosis and type 2 primary ketosis [2,7,9,10]. The recent years have seen a growing interest in the prediction of biological processes characterized by high complexity and non-linearity based on the use of artificial neural networks (ANNs). These networks provide tools that have been used successfully for the identification and modeling of processes observed in a range of scientific disciplines [17,18,19,20,21,22]. The high efficiency of ANNs and the high operating speed of neuron-based models imply a wide variety of applications oriented towards modeling and gaining insight into processes studied as part of animal husbandry [17,18,19,20,21]. The most popular and widely used type of unidirectional network is a multilayer perceptron (MLP) network combined with an error backpropagation algorithm. The error backpropagation method is based on the generalization of the so-called DELTA rule, designed for the purpose of multilayer neural networks [17,18,20]. MLP ANNs use “global approximation”, where each neuron has an impact on the results of mapping over the entire data space [17,23,24]. In MLP networks, any non-linear function can be modeled using a single hidden layer [25,26]. This eliminates the need for defining the network topology and the number of hidden layers. Another distinctive feature of MLP networks is their uncomplicated and straightforward network structure, and the network training process uses the method of unsupervised or supervised training [26], including the delta rule that provides the basis for a majority of supervised training methods [25,26,27,28]. The supervised training of unidirectional multilayer networks means that the form of an output vector (desired result) is precisely known for a given input vector. The purpose of the training process is to identify weighting factors that enable the achievement of network output values that are identical (or very similar) to the actual output values [17,23,26]. As a result, the network training process aims at minimizing the network error, defined as an aggregate measure of differences between the actual output values and those calculated by a network. The error is most commonly calculated as a sum of squared differences between the respective values [23], using the following formula:(1)$$E=\sum_{i=1}^{N}{\left(d_{1}–y_{1}\right)}^{2},$$

*d*_1_—actual input value

*y*_1_—value determined by network

*N*—size of training dataset

The training of neural networks is a multistage process where successive approximate values of parameters are determined during each stage of the process. The consecutive stages are referred to as training epochs [23,25]. An epoch is defined as a single cycle of an algorithm through the entire training dataset, and it includes a one-off presentation of all training cases and modifications of network parameters (weighting factors and threshold levels) implemented on that basis [23,25]. The input variables that are of immediate relevance to the network training process are selected based on a sensitivity analysis. The purpose of the analysis is to investigate the effects of the elimination of individual explanatory variables on the total network error [23,26,29]. Additionally, a Pearson’s correlation between the actual and the predicted value was calculated for each network as well as the error, defined as the sum of squared deviations between the input and the output values.

## 2. Materials and Methods

### 2.1. Selection and Sampling of Dairy Herds

The study material included information on dairy cows of the Polish black-and-white Holstein-Friesian breed. Cows included in the study originated from production farms located in South Poland (*n* = 3000) and had a calving date between 2016 and 2019. All cows were fed the same diet during the dry period and early lactation. Milk yield was recorded routinely by an automatic device installed in the milking parlor and lactation information came from a herd management program, Afimilk (Afifarm version 3.0, Afikim, Germania Dairy Automation, Waunakee, WI, USA). The animals were from different age groups and at different stages of lactation when milk samples were collected, which was between 5 and 60 days into the post-calving period. The data used for the calculations included milk composition as determined based on samples collected during test milkings and as received from the Polish Federation of Cattle Breeders and Milk Producers (PFHBiPM). Calculations were performed based on data for 1520 dairy cows, including daily milk yield (kg), percentage of fat (FP), protein (PP) and lactose (LAC), the content of urea (mg/L) (UR), acetone (mmol/L) (ACE) and β-hydroxybutyric acid (mmol/L) (BHB), and somatic cell count (thousand cells/mL) (SCC). All these parameters were used as input variables for all MLP network models. Table 1 shows the descriptive statistics of the initial dataset.

### 2.2. Laboratory Analysis

Blood samples were collected at approximately 20 min post-milking by puncture of the median caudal vein or artery or from the tail vein, using disposable evacuated blood collection tubes containing an anticoagulant (EDTA). Blood BHB levels from serum samples were measured using a spectrophotometer (model: UV-5100). The quantitative analysis of BHB by diphenyl carbazide absorptiometry was introduced. All cells were measured for the auto-zero values. As a result of the analysis, a calibration curve with a high coefficient (R^2^ = 0.9999) was obtained. The standard quartz cuvette (15 mm) was used, with a measurement wavelength of 540 nm. The reagent used was a Water quality test kit (Cr^6+^) with a range of quantitative analyses from 0.02 to 1.0 mh/L. The output variable value to be used for ANNs was determined based on a blood BHB level ≥ 1.2 mmol/L which is typical of subclinical ketosis [3,30]. A total of 168 cows had a blood BHB level in this range. The input variable was defined as the cow’s health status, i.e., a zero-one parameter: healthy cow = 0 and ketosis-affected cow = 1.

### 2.3. Approach

The study was conducted using an MLP ANN simulator as implemented in the STATISTICA12 statistical package (StatSoft). An MLP is formed by elementary processing units, the so-called neurons, and falls under the category of feedforward algorithms. The main components of the MLP model are the Summation function and the Activation function. The network training process was carried out using a training dataset and different numbers of neurons in the hidden layer (from 8 to 15) [16,24] as well as various activation functions in the hidden and output layers. A linear activation function was used for the network. The hidden layer was comprised of linear neurons, for which the aggregate input value was the sum of weight-ranked inputs, i.e., a scalar product of an input vector and a vector of weighting factors, i.e., a vector of neuron parameters (Table 1). In each repetition, after the weighted sums are forwarded through all layers, the gradient of the Mean Squared Error is computed across all input and output pairs. Then, to propagate it back, the weights of the hidden layer are updated with the value of the gradient. The activation functions used for the hidden and output layers are specified and characterized in Table 1. The perceptron training process was carried out using the error backpropagation method and an algorithm designed for supervised ANN training [23,26]. The Backpropagation process is a learning mechanism that allows the MLP to iteratively adjust the weights in the network, with the goal of minimizing the cost function.

#### 2.3.1. Data Preprocessing for Multi-Layer Perceptron

For the best performance of the feedforward, back propagation architecture process four combinations of activation functions for hidden and output layers were used (Table 2). The dataset was divided at random into three subsets: a training dataset (modification of weighting factors) containing 70% of the data, as well as a testing dataset (monitoring of the network training process) and a validation dataset (evaluation of networks following the training process), each containing 15% of the data. Back-propagation is only used during learning and training sets [28].

#### 2.3.2. Feature Selection

During the first stage of the network training process, the training and validation datasets were used, which made it possible to check the network training level. During the training of each network model, parameters of neurons, that is, the vector of weighting factors, were modified so as to obtain a model that described the relationship between output variables and the input variable as accurately as possible [Figure 1]. The modification of weighting factors was continued until the minimum approximation error was achieved. Neuron-based models also enabled ranking the pre-defined explanatory variables according to their importance based on the coefficient determined during the network sensitivity analysis. Variables were considered significant if the coefficient was greater than or equal to 1.0 [23,26]. The network training level was determined based on the validation dataset. The last stage of the network model training process involved network testing using the testing dataset composed of input variables that had not been used for the network training process up to that point. The error values were calculated as the sum of squared deviations between the pre-defined input value for a network model and the respective output value.

During the final stage of the network development process, sensitivity and specificity were calculated for the resulting network model. Sensitivity is the ratio of the number of true positive results to the total number of true positive (TP) and false negative (FN) results [31]. Sensitivity equal to 1.0 (or 100%) means that all ketosis-affected cows were correctly diagnosed. Specificity is the ratio of the number of true negative results to the total number of true negative (TN) and false positive (FP) results. Specificity equal to 1.0 (or 100%) means that all healthy cows were identified as such by the network. Results of sensitivity and specificity tests provided the basis for plotting receiver operating characteristic (ROC) curves that are useful for data analysis [31]. The optimum cut-off point was determined based on the ROC curve, which reflects the relationship between the percentage of true positive test results (sensitivity) and the percentage of false positive results (1—specificity). An optimum cut-off point is a point located on a ROC curve that is closest to the point having coordinates (0.1). This makes it possible to choose a threshold level for a diagnostic test that offers the optimum sensitivity and specificity of a network model. The generation of networks was initiated using eight input variables (milk components) and one output—binary variable (blood BHB level). For network models with different configurations of hidden and output activation functions and a different number of neurons in the hidden layer, the training process was run 100 times, resulting in a total of 168,000 network models. Out of the generated models, only those models showing the highest linear correlation coefficients were chosen for use during network training, testing and validation cycles. Based on sensitivity analysis, the input variables were selected that were the most relevant to the training of respective network models.

#### 2.3.3. Archiving Models

The network models were recorded using the Predictive Modeling Mark-up Language (PMML) that allows the use of a predictive model outside the related generation and testing environment. Therefore, it would be possible to effectively initiate and use the programmed network models in breeding and production practice regardless of the dataset used for their generation [32].

## 3. Results

A number of different activation functions were used in the hidden and output layers for the generated network models (Table 3). For a network model with 8 neurons in the hidden layer, the best network training results were achieved when a linear function was used in the hidden and output layers. A linear function was also used as an output activation function in models with 12, 13, 14 and 15 neurons in the hidden layer. Another function—an exponential function—was used as a hidden activation function in models with 9 and 14 neurons in the hidden layer. The hyperbolic tangent was used as a hidden activation function in models with 10, 12 and 13 neurons and as an output activation function in a model with 9 neurons. A sinusoidal function was used as a hidden activation function in a model with 15 neurons in the hidden layer and as an output activation function in models with 10 and 11 neurons in the hidden layer. Linear functions were used five times for the output activation and once for the hidden activation. Exponential functions were used for the hidden activation in two network models only. The hyperbolic tangent was used in three network models for the hidden activation and once as an output activation function. Sinus functions were used only once for the hidden activation and twice for the output activation of a network.

Table 4 shows network models characterized by the highest Pearson’s coefficients of linear correlation between the actual data (presence or absence of ketosis) and the value predicted by a network, and the lowest error value for a given model. The coefficients of correlation between the actual and the network predicted values in the training dataset ranged between 0.95 for a model with 11 neurons and 0.96 for models with 9, 12 and 15 neurons in the hidden layer. The capacity of a network to generalize relationships learned based on the training dataset can be measured using correlation coefficients in the validation and testing datasets. These coefficients in the validation dataset ranged between 0.64 for a network with 8 neurons and 0.66 for a network with 11 neurons in the hidden layer, and in the testing dataset, they ranged between 0.72 for networks with 13 and 14 neurons and 0.77 for a network with 11 neurons in the hidden layer. The training errors in the training dataset ranged between 0.50 for a network with 15 neurons and 0.96 for a network with 9 neurons in the hidden layer, in the validation dataset—between 0.56 (11 neurons) and 0.65 (8 neurons), and in the testing dataset—between 0.44 (13 neurons) and 0.49 (9 neurons in the hidden layer). Based on sensitivity analysis, the input variables were selected that had a significant impact on the effectiveness of a trained network. If the sensitivity coefficient for a variable was less than 1.0, the respective variable was removed from the input dataset prior to generating subsequent network models.

Table 5 shows the results of the sensitivity analysis of input variables according to the number of neurons in the hidden layer of the network. Out of the eight input variables, BHB and ACE levels were indicated as variables relevant to the training of all networks, regardless of the number of neurons. Additionally, for networks with 15 neurons in the hidden layer, the sensitivity analysis showed that FP was a significant variable in the training process while for networks with 14 neurons in the hidden layer, of relevance to the training process were such variables as FP, LAC and PP. Results of the analysis of sensitivity and specificity of the generated networks are shown in Table 6. The areas under the curve (AUCs) were similar and ranged between 0.82 for a network with 12 neurons and 0.89 for a network with 14 neurons in the hidden layer. The best sensitivity (0.84) was observed for a network with two input variables and 9 neurons in the hidden layer, however, the specificity of the network (0.61) was lowest among all compared networks. On the other hand, the greatest specificity (0.86) was determined for a network with five input variables and 14 neurons in the hidden layer. The network with 12 neurons in the hidden layer was characterized by a relatively low AUC, however, it showed a good combination of sensitivity (0.75) and specificity (0.82) for the optimum cut-off point at 0.52.

## 4. Discussion

There have been several reports of attempts to develop a simple and cost-effective method for the early identification of ketosis in cows [2,33,34,35]. These studies were mostly focused on monitoring changes in the composition of milk from cows diagnosed with ketosis on the basis of changes in ketone body levels in milk, such as BHB and ACE [14,33,34]. So far, artificial neural networks have not been used for this purpose, although a number of studies have involved attempts at using other statistical and mathematical tools in association with ROC curve analysis to define the incidence of ketosis in cows [17,33,35,36]. In Poland, the PFHBiPM has implemented a SYMLEK system procedure with a *K!* method, [37], that has been used since 2015 for the identification of cows at risk of ketosis. This method estimates the risk of subclinical ketosis in a cow using logistic regression with independent variables such as the content of BHB and ACE in milk from test milkings. The method enables the identification of cows at risk of ketosis with a sensitivity of 67–70% and relatively high specificity (85%). However, the *K!* method cannot be used for the real-time identification of cows at risk of ketosis. The current study involved the generation and training of an MLP artificial neural network designed for the identification of cows at risk of subclinical ketosis. Network models generated were characterized by similar or higher sensitivity and slightly lower specificity than the *K!* method. Networks with 9 neurons in the hidden layer were found to be the most effective and were characterized by a good combination of sensitivity (0.84) and specificity (0.61), and the AUC under the ROC curve (0.86). The MLP network model with 9 hidden neurons was the one for which the sensitivity analysis showed that BHB and ACE levels in milk were significant input variables in the network training process. However, a significant feature of the resulting network algorithm generated and tested as part of this study is the possibility of recording it using the predictive model markup language (PMML) which enables using the model for new data and continuous monitoring of animals in production farms in terms of the risk of subclinical ketosis. Carrier et al. (2004) achieved higher sensitivity (0.88) and specificity (0.90) in predicting the incidence of ketosis based on BHB levels in milk as determined using the Keto-Test milk strip, and the result was read according to the color scale. van Knegsel et al. (2010) reported low specificity (0.70) and sensitivity (0.80), comparable to those reported in this study, as determined using logistic regression based on the content of BHB and ACE in milk and blood, and evaluated using transform infrared spectrometry. It should be noted that in the latter study, cows were diagnosed with ketosis based on BHB and ACE levels, and the fat-to-protein ratio. On the other hand, Jorritsma et al. (1998) used the Ketolac milk test involving sticks that show a color change (formazan formation). Their prediction of the incidence of ketosis in cows, based on BHB levels in milk, achieved a relatively low sensitivity (0.40 and 0.62) and very high specificity (0.94 and 1.00), taking 0.2 mmol/L and 0.1 mmol/L as cut-off points for BHB levels in milk. Nielen et al. (1994) predicted the presence of ketosis in cows, based on the content of BHB in milk and urine (e.g., Acetest, bridgent test) and using Cohen’s kappa coefficient. The sensitivity resulting from their study, based on defining the threshold BHB level in milk in the range of 0.7 to 1.5 mmol/L, ranged widely between 0.22 and 0.90 while the specificity was between 0.96 and 0.99. Our considerations concerned the selection of a neural network model optimized with respect to the identification of individual cows with subclinical ketosis, i.e., a network characterized by both high sensitivity and high specificity. Taking into account all cows in a herd, their welfare and the production profitability, a method that provides the best possible sensitivity could be more useful, even at the expense of lower specificity. In such a case, the number of errors (identification of a smaller number of ketosis-affected cows as healthy and a greater number of healthy cows as ketosis-affected) will be less risky and costly for a breeder as compared to a situation when specificity is significantly higher than sensitivity [38]. If the sensitivity is higher than the specificity, a breeder may have to incur higher costs associated with the testing of the herd for ketosis, however, these costs would be nowhere near the costs of management of ketosis-affected cows that would otherwise be identified as healthy by an assessment with lower sensitivity [39].

The determination of sensitivity and specificity makes it easier to choose the final neural network models that are suitable for use in practical applications as a tool for the identification of cows at risk of ketosis in a herd. Both these parameters are important indicators of a method’s accuracy, and they can only adequately characterize a method’s efficiency when they are used in parallel.

## 5. Conclusions

Lately, farm labor has been identified as one of the major limitations to the further growth of the dairy industry. However, many dairy industries globally will have to face this challenge previously and so modern and precision technologies might also become essential tools in the dairy industry. The accessibility of advanced tools and innovative solutions for agricultural and stockbreeding operators will rely on the use of the high versatility of ANNs. In conclusion, predictions generated using MLP neural networks about the incidence of subclinical ketosis in cows based on milk parameters, that is: BHB, ACE, LAC and SCC, indicate how effective a specific model is, in comparison to results from other studies. Neural networks can help improve a model by modifying network design parameters in response to new data. Moreover, following the final selection of a specific neural network model, it is possible to immediately implement the resulting algorithm as a source code. The new precise IT tool is characterized by similar sensitivity and specificity to tools that are already in use, but with the advantage of offering continuous monitoring of cattle in the barn, e.g., as a complementary tool to animal breeding software, to help breeders identify cows at risk of subclinical ketosis.

## Figures and Tables

**Figure 1 animals-12-00332-f001:**
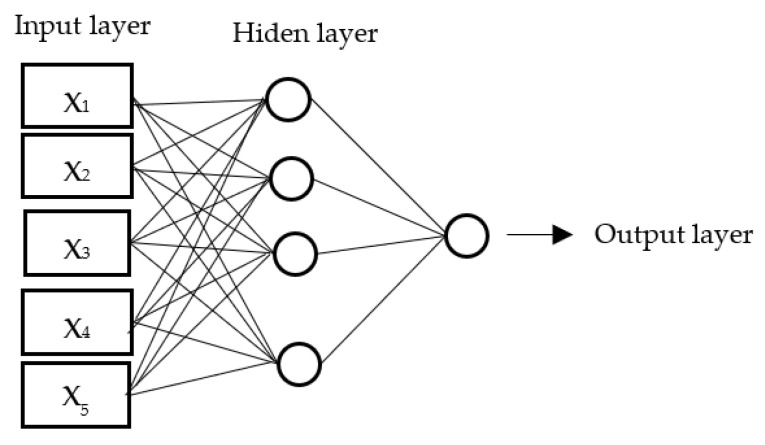
The Multilayer Perceptron structure.

**Table 1 animals-12-00332-t001:** Number of animals tested (*n* = 1520), mean and standard deviation of milk variables and β-hydroxybutyrate concentration (bBHB) in blood and milk (mBHB) according to lactation number.

Item	Lactation 1	Lactation 2	Lactation 3	Lactation ≥ 4
Number of cows	402	426	397	295
bBHB (mmol/L)	0.23 ± 0.33	0.65 ± 0.45	0.56 ± 0.39	0.85 ± 0.36
Milk variables				
Milk (kg)	26.89 ± 6.38	35.0 ± 0.69	32.80 ± 9.26	34.2 ± 10.1
Fat (%)	4.54 ± 1.00	4.5 ± 1.02	4.92 ± 1.03	4.68 ± 1.01
Protein (%)	3.24 ± 0.33	3.4 ± 0.38	3.30 ± 0.40	3.27 ± 0.34
Lactose (%)	4.85 ± 0.23	4.8 ± 0.20	4.76 ± 0.27	4.70 ± 0.24
Urea (mg/L)	197.56 ± 70.05	207 ± 77.14	203.12 ± 74.34	179.42 ± 73.66
SCC (1000/mL)	561.4 ± 1081.9	591.1 ± 1252.06	725.42 ± 1188.15	834.31 ± 1401.07
Acetone (mmol/L)	0.15 ± 0.18	0.1 ± 0.12	0.15 ± 0.17	0.13 ± 0.13
mBHB (mmol/L)	0.09 ± 0.13	0.1 ± 0.10	0.11 ± 0.11	0.86 ± 0.61

Number of cows, mean and standard deviation of blood β-hydroxybutyrate concentration (bBHB), milk yield, fat percentage, protein percentage, lactose percent, urea concentration, somatic cell score (SCC), acetone and milk β-hydroxybutyrate concentrations (mBHB).

**Table 2 animals-12-00332-t002:** The activation functions of the hidden and output layer used to train MLP network.

Type of Network	Type of Function	Function Model
MLP	Linear	*y = ax + b*
	Hyperbolic tangent	$y=tgh\left(\frac{\beta x}{2}\right)=\frac{1-e^{-\beta x}}{1+e^{-\beta x}}$
	Exponential	$f\left(x\right)=a^{x}\;\;,\;\;\;a>0$
	Logistic	$f\left(x\right)=\frac{1}{1+e^{-\beta x}}$
	Sinus	*f(x) = sin(x)*

MLP—multi-layer perceptron.

**Table 3 animals-12-00332-t003:** Activation function for chosen MLP networks.

ID MLP	Activation Functions	
	Hidden	Output
2-8-1	linear	linear
2-9-1	exponential	tangens
2-10-1	hyperbolic tangent	sinus
2-11-1	linear	sinus
2-12-1	hyperbolic tangent	linear
2-13-1	hyperbolic tangent	linear
5-14-1	exponential	linear
3-15-1	sinus	linear

MLP—identity models: 3; 5; 3—input variables, 10–15—hidden neurons, 1—output variable.

**Table 4 animals-12-00332-t004:** Pearson’s coefficients of linear correlation for learning, testing and validation sampling and errors for learning, network testing and validation sampling.

ID MLP	Coefficient Correlation			Error Function (SOS)		
	Training	Testing	Validation	Training Error	Testing Error	Validation Error
2-8-1	0.96	0.75	0.64	0.95	0.489	0.65
2-9-1	0.96	0.73	0.64	0.96	0.49	0.63
2-10-1	0.97	0.73	0.65	0.88	0.46	0.56
2-11-1	0.95	0.77	0.66	0.81	0.45	0.56
2-12-1	0.96	0.74	0.64	0.89	0.46	0.59
2-13-1	0.95	0.72	0.65	0.77	0.44	0.57
5-14-1	0.96	0.72	0.65	0.52	0.46	0.60
3-15-1	0.96	0.72	0.64	0.50	0.45	0.59

ID MLP—identity models.

**Table 5 animals-12-00332-t005:** Network sensitivity analysis for ketosis.

ID MLP	Input Variable				
	BHB	ACE	LAC	FP	PP
2-8-1	7.332	2.842	-	-	-
2-9-1	7.520	3.616	-	-	-
2-10-1	8.533	3.110	-	-	-
2-11-1	5.637	2.169	-	-	-
2-12-1	6.216	3.289	-	-	-
2-13-1	6.509	4.120	-	-	-
5-14-1	2.989	2.710	1.822	1.292	1.023
3-15-1	1.568	1.122	-	1.239	-

MLP identity model: 2; 5; 3—input variables, 8–15—neurons, 1—output variable; BHB—β-hydroxybutyric acid (mmol/L); ACE—acetone (mmol/L); LAC—lactose (%); FP—fat (%); PP—protein (%).

**Table 6 animals-12-00332-t006:** Diagnostic criteria of AUC under the ROC curve, sensitivity and specificity.

ID MLP	AUC ± SE	Cutoff	Sensitivity	Specificity
2-8-1	0.87 ± 0.01	0.46	0.63	0.83
2-9-1	0.86 ± 0.01	0.52	0.84	0.61
2-10-1	0.85 ± 0.01	0.49	0.72	0.81
2-11-1	0.84 ± 0.01	0.50	0.67	0.85
2-12-1	0.82 ± 0.01	0.52	0.75	0.82
2-13-1	0.85 ± 0.01	0.53	0.66	0.82
5-14-1	0.89 ± 0.01	0.54	0.67	0.86
3-15-1	0.85 ± 0.01	0.51	0.65	0.85

MLP identity: 2; 5; 3—input variables, 8–15—neurons, 1—output variable. AUC (Are Under Curve), ROC (Receiver Operating Characteristics), SE—standard error.

## Data Availability

None of the data were deposited in an official repository. Data may be available upon request by contacting the corresponding author.

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
