# Peer review of "The Use of Multilayer Perceptron Artificial Neural Networks to Detect Dairy Cows at Risk of Ketosis"

_animals, 2022, doi:10.3390/ani12030332_

Round 1

Reviewer 1 Report

in the attacment

Reviewer 2 Report

The study is interesting and it is of interest to develop new approaches for ketosis in dairy cows using different models. However, the data analysis quality is not ready enough for publication.
The paper does not provide enough of detail to judge the quality of the results. It is not clear how the optimization is done for the models parameters. It seems to be too complex models when it comes to the number of neurons. For such a small number of variables 2-5 having 8-15 neurons is too many. The error is hard to evaluate and should not be provided as well as Pearson correlation coefficient. These are the metrics for the regression analysis which was not done by ANN.
The authors should significantly improve the quality of the manuscript before it can be resubmitted.

Round 2

Reviewer 1 Report

suggestions: put as supplement tables with original data

optional: put in text key sentecies from author s text  - answer to reviewer.  It will be good and simple for reader of article
